# LEGATO: LARGE-SCALE END-TO-END GENERALIZABLE APPROACH TO TYPESET OMR

**Guang Yang**[1]    **Victoria Ebert**[1]    **Nazif Can Tamer**[1]
**Brian Siyuan Zheng**[1]    **Luiza Amador Pozzobon**[1]    **Noah A. Smith**[1,2]
[1]Paul G. Allen School of Computer Science & Engineering, University of Washington
[2]Allen Institute for AI
{gyang1,nasmith}@cs.washington.edu

## ABSTRACT

We propose Legato, a new end-to-end model for optical music recognition (OMR), a task of converting music score images to machine-readable documents. Legato is the first large-scale pretrained OMR model capable of recognizing full-page or multi-page typeset music scores and the first to generate documents in ABC notation, a concise, human-readable format for symbolic music. Bringing together a pretrained vision encoder with an ABC decoder trained on a dataset of more than 214K images, our model exhibits the strong ability to generalize across various typeset scores. We conduct comprehensive experiments on a range of datasets and metrics and demonstrate that Legato outperforms the previous state of the art. On our most realistic dataset, we see a 68% and 47.6% absolute error reduction on the standard metrics TEDn and OMR-NED, respectively.

## 1 INTRODUCTION

A substantial portion of written music exists only as photocopies of printed sheet music (e.g., IMSLP; Project Petrucci LLC, 2025). Digitalizing these images into modern, machine-readable formats would unlock data for a wide range of music analysis and synthesis applications, at an unprecedented scale. With this goal in mind, we tackle the problem of optical music recognition (OMR) to efficiently convert images of typeset scores to symbols.

The most successful approaches to this problem are end-to-end OMR systems (Ríos-Vila et al., 2024; Ríos-Vila et al., 2024; Ríos-Vila et al., 2023; Mayer et al., 2024; Calvo-Zaragoza & Rizo, 2018b), focusing exclusively on formats for piano, monophonic music, or single-system scores. More generalizable solutions require careful consideration of the diversity of *inputs* (i.e., complex layouts that contain multiple systems, staves and voices on a single page, as well as extensive text annotations such as titles and lyrics), *outputs* (each output format—e.g., MusicXML, ABC, **kern—has its own advantages for OMR and current evaluation heavily relies on output format choice), and *data* for training. Our main contributions are as follows.

- We construct a new multi-page large-scale OMR **dataset**, PDMX-Synth, rendered from symbolic scores in PDMX (Long et al., 2025; Xu et al., 2024) with diverse rendering schemes (§3).

- We apply a data-driven **tokenization** algorithm to symbolic music and find that it is capable of learning composite music concepts (§4.1).

- We introduce Legato, the first **end-to-end OMR model** built upon a pretrained vision encoder, and capable of recognizing multi-page typeset scores (§4).

- In comprehensive experiments, even those giving advantages to the baseline, we find that Legato achieves **state-of-the-art performance** on multiple OMR datasets, including a newly constructed sample from IMSLP (Project Petrucci LLC, 2025), with a large improvement over the previous best model (§6).

We release the code to reproduce our work at https://github.com/guang-yng/legato.

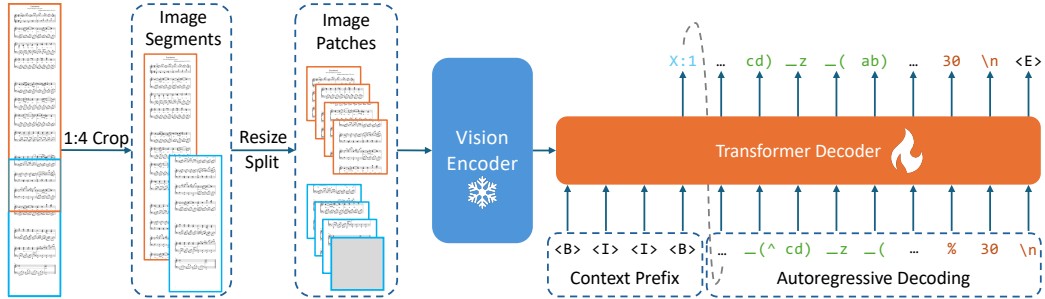

Figure 1: **Model architecture.** The input image is first cropped into overlapping segments with an aspect ratio of 1:4 or less, then resized and divided into four patches per segment (§4.2.1). The image patches are fed into a vision encoder (§4.2.2; parameters are frozen during training). The resulting latent embeddings serve as cross-attention keys and values in a transformer decoder, which autoregressively generates ABC tokens (§4.2.3). Special tokens ``, ``, and `<E>` denote `<|begin_of_abc|>`, `<|image|>`, and `<|end_of_abc|>`, respectively. For better visualization, here we use "_" to represent whitespace.

## 2 BASELINE AND DESIGN CONSIDERATIONS

### 2.1 MULTI-SYSTEM OMR

Our starting point is the success of Sheet Music Transformer++ (SMT++; Ríos-Vila et al., 2024), which is, to our knowledge, the only model designed to handle multiple systems in a score rather than single-staff or single-system scores (Mayer et al., 2024; Ríos-Vila et al., 2023). SMT++ is an encoder-decoder transformer model trained end-to-end on purely synthetic, full-page piano-form scores. It was trained on FP-GrandStaff (688 pages), which was generated by randomly concatenating single-system piano scores from GrandStaff (Ríos-Vila et al., 2023). SMT++ is the baseline against which we compare our approach.

Besides specialized models like SMT++, we also investigate how state-of-the-art, general-purpose multimodal models perform on OMR tasks. Specifically, we evaluate GPT-5 (OpenAI, 2025) under similar conditions as the other baseline.

### 2.2 OUTPUT SCORE REPRESENTATION

Many different symbolic score formats have been considered in past work on OMR; this paper considers **kern, ABC, and MusicXML.

**\*\*kern** is a musical representation designed within the Humdrum toolkit (Huron, 1997). \*\*kern is ASCII-based and places pitch and relative duration as the main focal points of the format, with visual information coming in second. \*\*kern explicitly models concurrent notes using spaces for notes in the same staff, and tabs for notes in other staves. It was designed with music researchers in mind, and with some study can be read by humans. \*\*kern is an OMR target in SMT++ (Ríos-Vila et al., 2024) and other systems (Ríos-Vila et al., 2024; Ríos-Vila et al., 2023).

The **ABC** music standard was introduced as an alternative ASCII-based form of musical notation (Walshaw, 2011). Similar to \*\*kern, it can be written and read by any text editor. The main attraction of ABC notation is the simple, concise format—a song that would take thousands of lines in other formats will take only tens in ABC, potentially reducing the computational cost for autoregressive models like our decoder. ABC notation can encode lyrics, title, tempo, decorations, articulations, and even some kinds of typesetting parameters, making it a nearly comprehensive format. The more explicit structure for musical engravings such as barlines and linebreaks, as well as a notation more similar to sheet music than that of \*\*kern, has led to ABC notation as a target for many systems (Wu et al., 2024; Wu & Sun, 2023; Casini et al., 2024), but to our knowledge, image-to-ABC OMR models have not yet been trained.

**MusicXML** is a tree-based music notation format introduced in 2001 (Good, 2001), widely adopted by commercial software such as MuseScore (MuseScore Ltd., 2021) and Sibelius (Avid Technology, 2025). Its popularity has made it a common target for OMR and music transcription tasks (Beyer & Dai, 2024), sometimes with simplifications (Mayer et al., 2024). However, compared to ABC and **kern, MusicXML is more verbose, harder to parse, and its hierarchical structure poses challenges for sequence models.

Each of these formats, despite their differences, carry much of the same information. They all include some intrinsic method of codifying visual score information such as barlines and phrasing, and each encodes note length as an absolute value, similar to typical human-readable sheet music—all elements that other popular symbolic music formats, such as MIDI, lack. This makes conversion among the three formats relatively simple, and many tools exist that can convert between formats (Vree et al., 2018; Cuthbert & Ariza, 2010; Cancino-Chacón et al., 2023; Pugin et al., 2014; Sapp, 2012), retaining core musical elements (notes, rests, measures). However, due to differences in the scope of information encoded across formats, certain elements—such as lyrics and articulations—may not be preserved during conversion.

We chose ABC as the output format for our model, as it is concise but nearly comprehensive and centers musical (rather than typesetting) elements. We believe the structure of ABC also lends itself well to usage with NLP techniques that help the model learn composite musical concepts, as we will see in the tokenizer (§4.1). Besides, we note that Markdown is widely used as the interface between general-purpose language models and humans, and some modern Markdown editors (e.g., StackEdit, JotterPad) are able to directly render scores from ABC code blocks, while **kern and MusicXML, to our knowledge, are not supported.

Finally, we focus on the recognition of musical notation rather than textual features of a musical score (e.g., lyrics), leaving the textual recognition to future work.

Given these design decisions and the strong starting point of SMT++, our approach to building a multi-system, multi-page, end-to-end OMR model includes constructing a large-scale dataset of score images with ABC-formatted representation (§3), and using it to train an encoder-decoder transformer model (§4) that builds on an existing pretrained image encoder.

## 3 DATASET

To train an end-to-end transformer-based OMR system, a large quantity of paired data (images with symbolic scores in the target format, ABC) is required.

### 3.1 A LARGE-SCALE OMR DATASET BASED ON PDMX

While the ABC notation project offers 750K examples available for free download, in genres from medieval music to pop music,[1] we found this data to be frequently monophonic. We believe these examples did not originate as full scores. We therefore turn to PDMX (Long et al., 2025; Xu et al., 2024), a dataset with 250K public domain MusicXML files collected from the online sharing forum MuseScore. We construct our ABC dataset, **PDMX-Synth**, by converting PDMX files from MusicXML to ABC format, and rendering images from a mixture of these two formats. During rendering, we exclude scores with aspect ratios greater than 10 (about 5% of the data). Training autoregressive models on long sequences is computationally expensive, and since such cases are rare, our approach sacrifices them for efficiency. Also, we canonicalize ABC format so that the model can more easily learn the format constraints, and also to simplify evaluation (§3.2). Due to filtering, conversion, and rendering loss, our final dataset contains 238,386 image-ABC pairs, about 93.8% of the original PDMX dataset.

### 3.2 CANONICAL ABC REPRESENTATION

We use the xml2abc script (Vree et al., 2018) to convert from MusicXML to ABC in batch mode. The following rules are applied to nearly-canonicalize ABC:[2]

---

[1] https://abcnotation.com/

[2] More could be done; e.g., voice numbering can still be swapped.

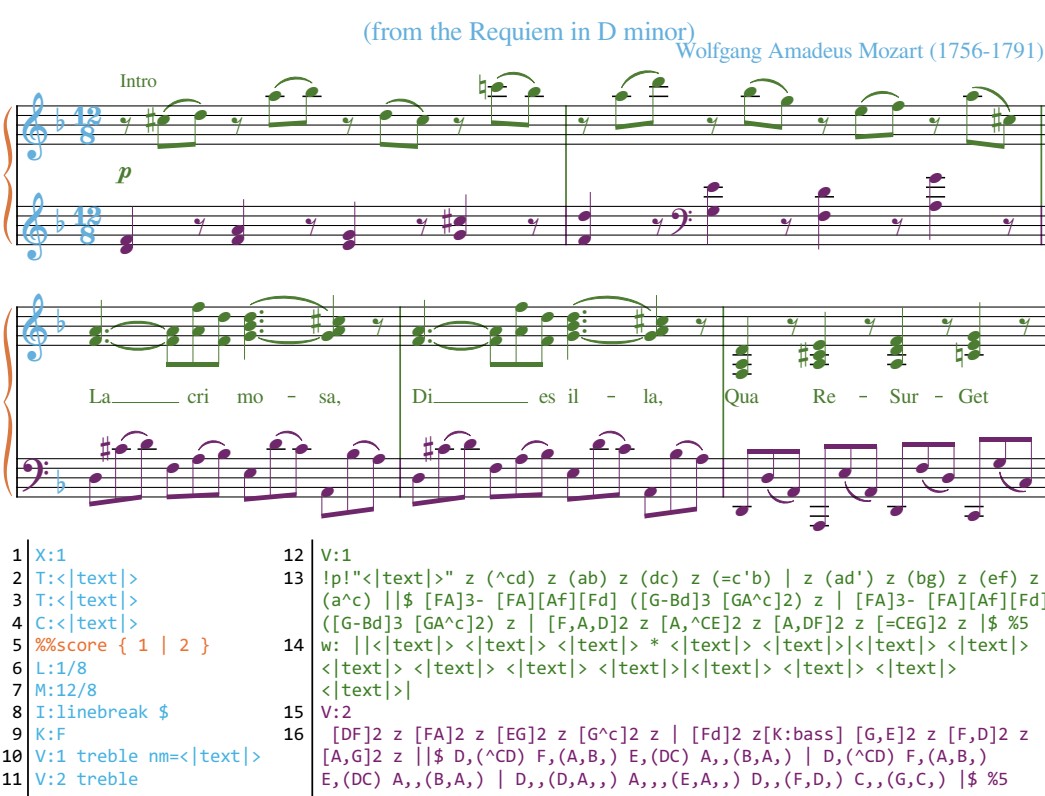

```
 1 │ X:1                              12 │ V:1
 2 │ T:<|text|>                       13 │ !p!"<|text|>" z (^cd) z (ab) z (dc) z (=c'b) | z (ad') z (bg) z (ef) z
 3 │ T:<|text|>                          │ (a^c) ||$ [FA]3- [FA][Af][Fd] ([G-Bd]3 [GA^c]2) z | [FA]3- [FA][Af][Fd]
 4 │ C:<|text|>                          │ ([G-Bd]3 [GA^c]2) z | [F,A,D]2 z [A,^CE]2 z [A,DF]2 z [=CEG]2 z |$ %5
 5 │ %%score { 1 | 2 }                14 │ w: ||<|text|> <|text|> <|text|> * <|text|> <|text|>|<|text|> <|text|>
 6 │ L:1/8                               │ <|text|> <|text|> <|text|> <|text|>|<|text|> <|text|> <|text|>
 7 │ M:12/8                              │ <|text|>|
 8 │ I:linebreak $                    15 │ V:2
 9 │ K:F                              16 │ [DF]2 z [FA]2 z [EG]2 z [G^c]2 z | [Fd]2 z[K:bass] [G,E]2 z [F,D]2 z
10 │ V:1 treble nm=<|text|>              │ [A,G]2 z ||$ D,(^CD) F,(A,B,) E,(DC) A,,(B,A,) | D,(^CD) F,(A,B,)
11 │ V:2 treble                          │ E,(DC) A,,(B,A,) | D,,(D,A,,) A,,,(E,A,,) D,,(F,D,) C,,(G,C,) |$ %5
```

Figure 2: An example of our canonical ABC representation (below) with a MusicXML-rendered image (above).

- **Transcribe real line breaks with $.** Line breaks are optional in ABC; explicitly marking them allows recovering original line breaks.

- **Force ABC files to break lines every 5 bars.** Textual line breaks in ABC have no semantics for scores. We enforce a fixed line length of 5 bars in the ABC text, except when fewer than 5 bars remain at the end of the score and retrain converter-generated comments `%[number_of_total_measures]` at the end of each line.

- **Fix the unit note to an 8th note.** The ABC grammar allows customized unit note length with `L:1/1, L:1/2, L:1/32, . . .`, so the same score could be transcribed differently with different unit note lengths. To simplify learning, we set `L:1/8`. This does not change the expressive power of the representation.

Since PDMX-Synth is designed for the OMR task, we replace all text contents in ABC representation with special `<|text|>` tokens. This includes the titles, instrument names, lyrics, and annotation contents quoted in the ABC tune body. Figure 2 shows an example of our canonical ABC representation.

### 3.3 Score Rendering

To construct an OMR training dataset from MusicXML or ABC files, it is important to choose an appropriate score renderer. It should be able to faithfully generate score images and represent most information contained in MusicXML files (e.g., fonts and spacing). Moreover, the rendered images should be varied enough so that the trained model will not overfit the default renderer parameters.

The FP-GrandStaff dataset used to train SMT++ (Ríos-Vila et al., 2024) uses the Verovio tool (Pugin et al., 2014) as the default renderer, and generates the images from **kern format. Since the **kern

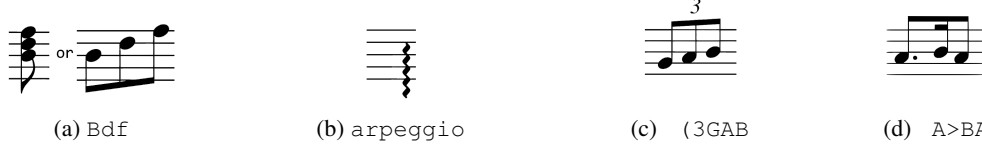

(a) `Bdf`    (b) `arpeggio`    (c) `(3GAB`    (d) `A>BA`

Figure 3: Example vocabulary items from tokenization.

format they use does not encode any typesetting information, the software's default rendering parameters are used, heavily limiting dataset diversity. To address this issue, we generate images with two rendering pipelines:

(i) *MuseScore 3.6.2* (MuseScore Ltd., 2021), which takes in MusicXML and outputs PNG.

(ii) *abcm2ps 8.14.15* (Moine, 2024), which takes in ABC and outputs SVG files. We further use CairoSVG 2.7.1 (CourtBouillon, 2023) to convert SVG files into PNG files.

To prevent default rendering parameters from prevailing in the dataset, we apply these visual augmentations:

- *For (i)*, the final images are augmented with **randomized image resolution** (i.e., randomly set the resolution parameter in MuseScore) and **randomized margin cropping** sampled from a uniform distribution. When cropping, the main score remains untouched.

- *For (ii)*, since ABC formats carry less typesetting information, more augmentations are applied:
    1. render a score with a single image or multiple images concatenated;
    2. with a probability of 50%, render the images in landscape mode;
    3. with a probability of 70%, add the measure numbers with different numbering styles;
    4. uniformly set the left and right margins;
    5. randomly scale the image by a fraction in $[0.9, 1]$.

Besides, for both renderers, the background color is sampled uniformly from the grayscale range $[192, 255]$. We release this synthesized ABC OMR dataset, PDMX-Synth, to support future research.

## 4 END-TO-END MODEL

Our model, Legato, follows the architecture of multimodal Llama (AI@Meta, 2024). As shown in Figure 1, the main components of our model are a pretrained vision encoder and a transformer decoder into ABC. The input score image is divided into segments, resized, and further split into four patches per segment, which are encoded into latent embeddings by the vision encoder. These embeddings are then used by the transformer decoder to autoregressively generate tokens in ABC format. We discuss our ABC tokenization method in §4.1, then turn to the architecture: details of image processing (§4.2.1), the use of a pretrained vision encoder (§4.2.2), and the transformer decoder (§4.2.3).

### 4.1 TOKENIZATION

Tokenization schemes for language models split a stream of characters into tokens from a fixed vocabulary. They can be based on an expert-defined vocabulary, as done with SMT++, whose vocabulary contains all possible **kern symbols (Ríos-Vila et al., 2024), or constructed in a data-driven fashion. We take the latter approach, which allows composite musical concepts like chords to be represented directly in the vocabulary if they are sufficiently frequent.

We adopt the byte-pair encoding method (BPE; Sennrich et al., 2016) for learning a tokenizer, widely used in natural language processing research and known for effectively capturing diverse patterns within a limited vocabulary, particularly when trained on large corpora. We choose to apply BPE

tokenization directly to the ABC representation of PDMX-Synth training set, with a vocabulary size of 4097, ensuring efficient representation and facilitating better model performance.

We find that our tokenizer captures some composite musical concepts like chords and short melodic phrases. For example, the C major triad, represented as CEG, emerges as a discrete token. This token exhibits contextual flexibility: within square brackets ([]), it denotes a simultaneous chord, whereas in the absence of brackets, it represents an arpeggiated sequence. When combined with duration tokens such as 2 or 4, CEG can represent a C major triad composed of quarter notes or half notes, respectively. More examples are shown in Figure 3.

## 4.2 MODEL ARCHITECTURE

Multimodal Llama effectively handles images with varying aspect ratios without significant distortion (AI@Meta, 2024). Given the complexity of the OMR task and the associated computational cost, we use the pretrained vision encoder from meta-llama/Llama-3.2-11B-Vision (836M parameters, frozen in our system) and train a 101M-parameter transformer decoder from scratch on ABC representation, along with a 5.9M-parameter linear multimodal projector to bridge between them. We refer to this model as Legato.

To control for model size when comparing with previous state-of-the-art methods, we also introduce a smaller variant, Legato$_{small}$, which has a 8.5M parameters in the decoder and 2.5M in the projector. This design has a comparable number of trainable parameters to SMT++, though the pretrained and frozen vision encoder adds substantially more.

### 4.2.1 IMAGE PROCESSING

Our input score image $I$ consists of the full score of a composition. Since a composition of music can be very long, the image $I$ might have a very large height, but the width stays relatively fixed (since scores are printed in portrait or landscape mode on standard paper sizes). We divided the image $I$ into multiple segments, each with an aspect ratio of 1:4 or less. Adjacent segments also have an overlap, ensuring that each segment retains contextual information.

We follow the approach used in multimodal Llama (AI@Meta, 2024) to further process each image segment. After resizing and cropping, each segment is divided into four smaller image patches. Therefore, starting from the original image $I$, we get a tensor $\mathbf{p} \in \mathbb{R}^{S \times 4 \times C \times D \times D}$, where $S$ is the number of segments, $C = 3$ is the number of color channels, and $D = 448$ is the internal image size.

### 4.2.2 VISION ENCODER

The Llama vision encoder was pre-trained on general-purpose images. It maps an image segment to an embedding. We conjecture that a vision encoder trained on diverse image data provides a strong starting point for OMR on score images, so we keep the vision encoder frozen in our training and testing experiments. Finetuning this module specifically for scores is a promising direction for future work.

We refer the reader to AI@Meta (2024) for details on the encoder, noting only that it provides an output embedding in $\mathbb{R}^{S \times 4 \times L \times 6d_v}$, where $L$ is the sequence length (specifically $1 + \left(\frac{D}{14}\right)^2$) and $d_v$ is the internal visual embedding dimension. In the Llama checkpoint used, $L = 1025, d_v = 1280$.

### 4.2.3 TRANSFORMER DECODER

We adopt a decoder architecture similar to that of multimodal Llama, but with a smaller scale. A linear projection is first applied to the latent embedding to match the decoder's hidden dimension $d_l$, enabling its use in cross-attention. The core of the decoder consists of a $L_d$-layer transformer, where cross-attention is selectively applied at a subset of layers denoted by $\Gamma_l$, while the remaining layers use only self-attention to reduce computational cost. The MLP module in each layer first upscales the dimension to $d_u$ then downscales back to $d_l$.

As illustrated in Figure 1, the decoder takes a sequence of tokens as input. It is trained to predict the next token in the sequence, and during inference, it autoregressively generates tokens one at a

time. In Legato, $d_l = 768$, $d_u = 1526$, $L_d = 18$ and $\Gamma_l = \{3, 7, 11, 15\}$. In Legato$_{\text{small}}$, $d_l = 320$, $d_u = 448$, $L_d = 8$ and $\Gamma_l = \{3, 5, 7\}$.

## 4.3 Pretraining Details

Both Legato and Legato$_{\text{small}}$ are trained for 10 epochs using a batch size of 32 and a learning rate of 0.0003. Following standard language model practices, we use the AdamW optimizer ($\beta_1 = 0.9$, $\beta_2 = 0.99$, $\epsilon = 10^{-6}$), a linear learning rate scheduler, and a warm-up ratio of 0.03. To improve efficiency, text sequences are truncated to 4096 tokens, and bfloat16 precision is used.

Due to the high cost of inference and metric computation, evaluation is performed every 5000 steps on a subset of 800 validation samples. The checkpoint with the lowest symbol error rate (SER) on ABC is retained: 65,000 steps for Legato and 60,000 steps for Legato$_{\text{small}}$.

## 5 Evaluation Metrics

Evaluation of the end-to-end OMR task has not yet converged on a single standard. Previous work (Ríos-Vila et al., 2024; Ríos-Vila et al., 2024; Mayer et al., 2024; Calvo-Zaragoza & Rizo, 2018a;b) report error rates in different score formats as there is no unified output format that is most suitable for this task. Typically, the chosen format is the one the proposed model was trained on. This choice might unfairly penalize the performance of models with a different output format and therefore hinder cross-model evaluation, as the conversion among formats is not always successful, and conversions of ill-formed outputs are undefined.

We adopt MusicXML as a unifying evaluation format across different models as it is able to carry all information required to typeset music scores. It is also not the format used to train Legato or SMT++, which yields a more fair comparison. Moreover, MusicXML is widely supported across music software, making it a reasonable final target for use-cases leading to editing of scores by humans. When necessary, we also provide error rates in **kern and ABC formats, which are the formats used in training SMT++ (Ríos-Vila et al., 2024) and Legato, respectively. Currently, both TEDn and OMR-NED evaluate textual elements. This means the final error rates include the edit distance between Legato's predicted `<|text|>` token and the actual text in the ground truth.

### 5.1 Tree Edit Distance with Note Flattening (TEDn)

Hajič Jr. et al. (2016) investigate different evaluation metrics on MusicXML and reach the conclusion that tree edit distance with `<note>` flattening is the best match to human evaluation. To elaborate, this metric first flattens all `<note>` elements in both XML trees and then uses normalized tree edit distance (edit distance between two trees divided by edit distance to recover the gold tree) as the final non-negative score. Flattening is used to decrease the high cost to delete a `<note>` element as it always contains many child elements. We use the implementation from Mayer et al. (2024). The most efficient edit distance algorithm requires $O(m^2 n^2)$ time, where $m$ and $n$ are the number of nodes in the two trees (Zhang & Shasha, 1989). Because of this cost, we truncate MusicXML files so that $m, n < 6000$ and use format-specific error rates to validate models during training. This truncation prevents the TEDn from capturing all errors.

### 5.2 Format-Specific Error Rates

A group of more widely-used metrics in OMR are character, symbol, and line error rates (CER, SER, and LER). They measure how much effort it takes to correct the predicted content, but they simplify the scores to text sequences rather than structured encoding and therefore fail to capture the magnitude of structural errors. These metrics are also relatively cheap, with $O(mn)$ runtime (for string lengths $m$ and $n$). Previous work (Ríos-Vila et al., 2024) uses these metrics on **kern, but our model is trained on ABC. Error rates on these two formats are not comparable since characters, symbols, and lines in these formats represent different concepts. So, we use converters to achieve an apples-to-apples comparison in both formats. However, these metrics still favor the model trained in the target format, as failed conversions for the other model result in empty outputs.

## 5.3 OMR Normalized Edit Distance (OMR-NED)

OMR-NED (Martinez-Sevilla et al., 2025a) is a novel evaluation metric for OMR that achieves a balance between computational efficiency and evaluation granularity. It is based on the sequence error rate computed between two sequences of musical measures, where the cost of transforming one measure into another is defined by the set edit distance between their constituent symbols. Insertion and deletion costs are specified for different categories of music symbols, enabling fine-grained assessment of model performance. This metric is both efficient and perceptually meaningful, with a time complexity of $O(M^2 S \log S)$, where $M$ denotes the number of measures and $S$ the average number of symbols per measure. Furthermore, OMR-NED is format-agnostic, provided that a robust parser is available to extract symbolic representations from the model's output.

The current OMR-NED implementation is optimized for **kern through syntax correction, while ABC v2.1 lacks parser support; thus, ABC outputs are converted to MusicXML for symbol extraction, which may introduce errors and disadvantage ABC.

## 6 Experimental Evaluation and Results

One limitation of many prior end-to-end OMR models is that they are only evaluated on the test split of the same datasets used for training (Ríos-Vila et al., 2024; Ríos-Vila et al., 2023; Mayer et al., 2024), or continue training on evaluation datasets (Ríos-Vila et al., 2024), which raises the risk of overfitting to a narrow population of scores. General OMR inputs exhibit significant variability, while individual datasets often stem from limited sources. Such restricted datasets are easier to overfit, artificially boosting evaluation metrics. We therefore evaluate Legato and the SMT++ baseline on a diverse set of OMR datasets, none of which are used for training or validation, ensuring a more robust and unbiased assessment of generalization performance. These include: (i) OpenScore String Quartets (Gotham et al., 2023), using both realistic and rendered images, (ii) OpenScore String Quartets violin parts, using Verovio-rendered images, (iii) OpenScore Lieder (Gotham & Jonas, 2022), using both realistic images and newly rendered ones, and (iv) newly manually converted piano scores from IMSLP. Furthermore, we also evaluate Legato on JAZZMUS (Martinez-Sevilla et al., 2025b), a handwritten score dataset. As this dataset deviates from Legato's primary training distribution, the model demonstrates limited generalization to handwritten notation.

To provide a reference of current general VLM's ability, we also evaluate GPT-5's performance on these datasets. Results are shown in Table 1. GPT-5 consistently outperforms SMT++ on TEDn but underperforms on OMR-NED. This is likely due to **kern syntax correction made by OMR-NED.

All evaluations for Legato are performed with beam search (beam size 10, max length 2048) and repetition penalty of 1.1, except PDMX-Synth, where we use beam size of 3 due to very large images. The baseline settings for both SMT++ and GPT-5 are detailed in appendix §A.

**Evaluation on PDMX-Synth.** We first evaluate the Legato models on 800 items from the PDMX-Synth test split. This evaluation establishes the quality of the Legato models' "in-domain" ability and on the output format they were trained to produce (ABC). Legato achieves (ABC) character, symbol and line error rates of 23.3%, 25.8%, and 31.7%, respectively, while Legato$_{small}$ achieves 36.4%, 39.2%, and 45.9%, respectively. We also evaluate the multi-page performance (appendix §B.6).

**Evaluation on OpenScore String Quartets.** To compare fairly with SMT++, we use a third-party dataset, OpenScore String Quartets (Gotham et al., 2023), mainly from the 19th century; this data was not used to train either model. The dataset provides MusicXML files, and for some entries, a scanned PDF is also available, from which the MusicXML was annotated. We extract a subset of the OpenScore String Quartets dataset containing scanned images of real scores, and render the corresponding clean images from the associated MusicXML files. We call these two different kinds of images "Camera" and "Rendered" and evaluate on both. The results are shown in Table 1 (blocks 1–2). Note that for both SMT++ and Legato, the output is not guaranteed to be convertible to MusicXML, a necessary step for calculating the TEDn evaluation metric. However, even when favoring SMT++ by evaluating only on instances where it produces valid results (rows marked "TEDn$_{convert}$"), Legato still outperforms it. Additional experiments further reducing the differences between these models are presented in appendix §B.5.

Table 1: **Experimental results on various datasets and metrics.** Lower is better for all metrics. $n$ represents the number of samples tested for the metric. TEDn is the primary metric, requiring outputs to be converted to MusicXML. TEDn$_{convert}$: evaluated only on instances where SMT++ produces outputs that can be successfully converted to MusicXML; Legato outputs always converted to MusicXML successfully. OMR-NED is format agnostic, built on extraction of symbols from any format. For OMR-NED, **kern outputs are automatically corrected for syntax errors, while ABC is first converted to MusicXML (again, always successful in practice) before symbol extraction. OpenScore String Quartets is the most challenging dataset, since it has much denser score images. All metrics are explained in §5.

| Metric | $n$ | GPT-5 | SMT++ | Legato | Legato$_{small}$ |
|---|---|---|---|---|---|
| *1. Camera OpenScore String Quartets (252 pages)* | | | | | |
| TEDn | 252 | 90.5 | 98.6 | **60.4** | 84.1 |
| TEDn$_{convert}$ | 18 | 93.1 | 80.2 | **58.6** | 84.1 |
| OMR-NED | 252 | 97.6 | 94.7 | **58.2** | 93.5 |
| *2. Rendered OpenScore String Quartets (252 pages)* | | | | | |
| TEDn | 252 | 93.0 | 97.9 | **52.1** | 78.4 |
| TEDn$_{convert}$ | 28 | 93.8 | 81.3 | **50.5** | 78.9 |
| OMR-NED | 252 | 97.8 | 94.3 | **32.9** | 88.5 |
| *3. Camera OpenScore Lieder (64 pages)* | | | | | |
| TEDn | 64 | 91.4 | 98.7 | **47.0** | 82.7 |
| TEDn$_{convert}$ | 6 | 98.6 | 86.3 | **44.5** | 57.2 |
| OMR-NED | 64 | 97.0 | 86.4 | **55.8** | 85.1 |
| *4. Rendered OpenScore Lieder (64 pages)* | | | | | |
| TEDn | 64 | 91.0 | 97.4 | **26.5** | 68.8 |
| TEDn$_{convert}$ | 4 | 97.5 | 58.1 | **8.7** | 52.6 |
| OMR-NED | 64 | 97.2 | 78.0 | **34.3** | 69.4 |
| *5. IMSLP Piano Scores (32 pages)* | | | | | |
| TEDn | 32 | 96.7 | 97.7 | **29.7** | 76.9 |
| TEDn$_{convert}$ | 3 | 90.3 | 75.2 | **3.8** | 43.7 |
| OMR-NED | 32 | 98.7 | 91.9 | **44.3** | 86.7 |

**Evaluation on OpenScore Lieder.** To enable a more comprehensive evaluation, we assess both models on the OpenScore Lieder dataset (Gotham & Jonas, 2022), which contains songs by 19th-century composers. Similar to OpenScore String Quartets, we obtain the source PDFs to generate both camera and rendered images. Again favoring SMT++, we retain only the piano part by masking the vocal staff with white boxes. This approach is similar to the one used in the OLiMPiC dataset (Mayer et al., 2024), although we use full-page images instead of single-system excerpts. As shown in Table 1 (blocks 3–4), both Legato variants outperform SMT++ by a large margin.

**Evaluation on IMSLP Piano Scores.** Masking the vocal staff with white boxes still introduces artifacts such as large gaps between staves. For a more realistic evaluation on piano-form camera scores, we manually annotated 32 full-page piano scores from IMSLP (Project Petrucci LLC, 2025). We ensure there is no overlap with PDMX-Synth and it includes only scans produced before the adoption of modern typesetting software such as MuseScore or Verovio. This allows us to eliminate biases introduced by synthetic typesetting and data source overlap. This small evaluation dataset is publicly available in our codebase. The results are presented in Table 1 (block 5).

Although the dataset is relatively small, it provides a fair and realistic comparison—both models are designed to recognize piano scores, and the image sources reflect real-world scenarios, as many piano learners obtain their scores from IMSLP. We show some examples for comparing the outputs in the appendix §B.7.

## 7 CONCLUSION

In this work, we propose Legato: a state-of-the-art, end-to-end generalizable model for typeset OMR. Legato is capable of recognizing multi-page realistic typeset score images and outputs ABC

representations. To achieve this, we leveraged a pretrained vision encoder (frozen), and trained a tokenizer and a decoder model on more than 214K samples from PDMX-Synth, a processed version of the PDMX dataset.

Legato achieves state-of-the-art performance on all evaluated datasets, even when favoring previous methods, and it represents a significant step forward as the first multi-page end-to-end OMR model for typeset scores. As future work, researchers can investigate how to finetune modern vision encoders to further adapt to the specific challenges of OMR.

ETHICS STATEMENT

This work has made use of large language model (LLM) tools, including experiments on GPT-5, as well as assistance with language polishing and the generation of acronyms. The conception of this paper, together with the complete codebase, PDMX-Synth dataset, and all experiments related to Legato, are original contributions of the authors.

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

## A  BASELINE SETTINGS

### A.1  SMT++

There are two available checkpoints for SMT++: `antoniorv6/smtpp_mozarteum` and `antoniorv6/smtpp_polish_scores`. For our evaluation, we use the checkpoint `antoniorv6/smtpp_kmozarteum` because its training data, the Mozarteum dataset (Rink, 2021), is more similar to our evaluation dataset. This choice is favorable toward SMT++.

SMT++(Ríos-Vila et al., 2024) claimed "we used a 5-fold cross-validation" for evaluation on *Mozarteum* and *Polish Digital Scores*. This is not a realistic evaluation since it only shows the capability of the model to overfit the distribution of these two datasets, and it's not the performance of a pre-trained model on real-world data. We do not know which portion of the dataset their released checkpoints were trained on.

For generation hyperparameters, we use the default setting from SMT++'s repository `https://github.com/antoniorv6/SMT-plusplus` and set max generation length to 2048.

### A.2  GPT-5

Legato's performance was compared against the `gpt-5-2025-08-07` of GPT-5. Due to the full-page setting, we found that without additional prompting, GPT-5 would often avoid answering and resort to outputs such as

```
The notation in the provided image is not legible enough to transcribe
    accurately. Please upload a higher-resolution image or a closer crop
    so the notes, rhythms, and accidentals can be read.
```

Therefore, we added the following system prompt to make sure GPT-5 always outputted valid ABC.

```
You will be given an image of a sheet of music.
Transcribe it into valid ABC 2.1 notation. Try your best to transcribe
    and make a reasonable guess if the image is not clear.
Output only the ABC (no explanations), preferably inside a single ```abc
    fenced block.
IMPORTANT: NEVER give outputs like: 'Unable to transcribe from the
    provided image due to insufficient resolution/clarity.' If you can't
    tell, give your best guess.
```

```
!!!ALWAYS OUTPUT VALID ABC, DO NOT GIVE ANY ENGLISH OUTPUT. GIVE YOUR
    BEST GUESS IF YOU ARE NOT SURE!!!
```

This is the default setting and is used for evaluation in Table 1. For more optimization and evaluation on general-purpose VLMs, we refer readers to appendix §B.4.

# B ADDITIONAL EXPERIMENTAL RESULTS

## B.1 ABLATION STUDY

To further quantify the improvements brought by different design consideration of Legato, we conduct an ablation study on IMSLP Piano Scores. In addition to the original SMT++ and Legato as shown in Table 1, we train the following models:

- SMT++ on ABC Full-Page GrandStaff
- SMT++$_{large}$ on ABC Full-Page GrandStaff
- SMT++$_{large}$ on PDMX-Synth

SMT++$_{large}$ is an enlarged version of SMT++ with in total 177M trainable parameters. In contrast, Legato has 107M trainable parameters. For the dataset, we converted the **kern-based Full-Page GrandStaff used by SMT++ into ABC, and follow the exact same training paradigm, including syetem-level pre-training and curriculum fine-tuning. We use the same training hyperparameters for SMT++, and for SMT++$_{large}$ we pick the best learning rate from $\{1e-4, 3e-5, 1e-5, 3e-6\}$. For generation, we follow the setting of SMT++. The evaluation results are shown in Table 2.

Table 2: **Ablation study on IMSLP Piano Scores.** Lower is better for all metrics. All metrics are explained in §5. †: Following SMT++ style of data processing, i.e., syetem-level pre-training and curriculum fine-tuning.

| Architecture
Training Dataset
Format & Tokenizer | SMT++
GrandStaff†
**kern | SMT++
GrandStaff†
ABC | SMT++$_{large}$
GrandStaff†
ABC | SMT++$_{large}$
PDMX-Synth
ABC | Legato
PDMX-Synth
ABC |
|---|---|---|---|---|---|
| *IMSLP Piano Scores (32 pages)* | | | | | |
| TEDn | 96.7 | 74.6 | 73.0 | 80.5 | 29.7 |
| OMR-NED | 91.9 | 70.9 | 82.9 | 96.2 | 44.3 |

Comparing the two columns of SMT++ on GrandStaff of **kern and ABC, we observe a clear improvement. This improvement is brought by the usage of ABC format and our BPE tokenizer, verifying that ABC + BPE tokenization is a practical setting for training auto-regressive models for OMR.

We then discovered that SMT++ failed to scale — there is no improvement by either enlarging the model size or data size, even though we carefully tuned the hyperparameters. We think that SMT++$_{large}$ has too many parameters and is easy to overfit to the artifacts in the training data, as we observe a low validation loss but high testing errors. Legato, on the other hand, works on the robust features encoded by the pre-trained vision encoder, and thus is more generalizable to the test data, even if only trained on synthetic scores. As a conclusion, the improvement arises from the synergy between the Legato pre-trained encoder, the Legato architecture, and the larger, more diverse PDMX-Synth dataset.

## B.2 LEARNING CURVE OF LEGATO

To validate the effect of data scaling on our model's performance, we present the training curves of Legato in Figure 4. The curves show a consistent decrease in validation loss and error rates as the number of training steps increases, indicating effective learning and improved generalization. The metrics eventually plateau around 55K training steps, suggesting that Legato has largely converged by this point and that additional training yields diminishing returns.

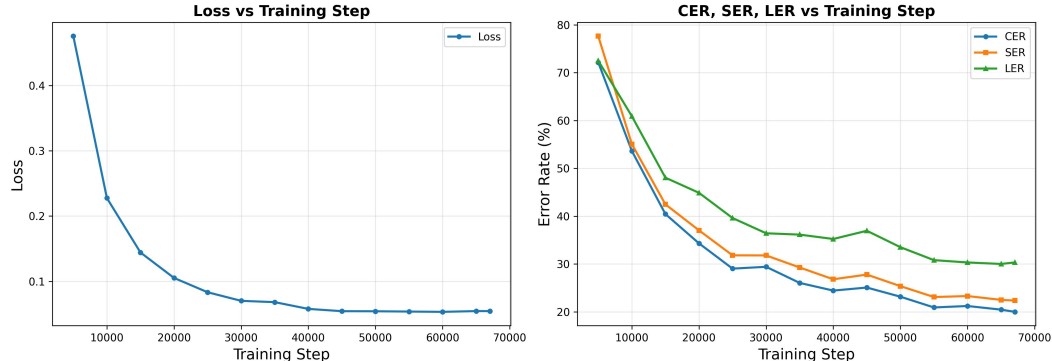

Figure 4: **Learning Curve of Legato.** The validation loss and error rates are tested on the validation split of PDMX-Synth.

## B.3 CONVERSION FAILURE RATES

We show in Table 3 the conversion failure rates of **kern to MusicXML on SMT++'s outputs. Notice that all outputs of Legato can be successfully converted to MusicXML.

Table 3: **Failure rates of **kern-to-MusicXML conversion on SMT++'s outputs.**

| Dataset | Total Files | Failed Files | Failure Rate |
|---|---|---|---|
| *Camera OpenScore String Quartets* | 252 | 234 | 92.9% |
| *Rendered OpenScore String Quartets* | 252 | 224 | 88.9% |
| *Camera OpenScore Lieder* | 64 | 58 | 90.6% |
| *Rendered OpenScore Lieder* | 64 | 60 | 93.8% |
| *IMSLP Piano Scores* | 32 | 29 | 90.6% |

We observe that the failure rates for **kern-to-MusicXML conversion are very high. This is primarily because most outputs from SMT++ are not well-formed. Moreover, the **kern format is highly susceptible to errors — for example, the omission of a tab character can make it difficult to repair the output. The ABC format, on the other hand, is more robust to such errors.

## B.4 RESULTS ON GENERAL-PURPOSE VLMS

To further investigate the capability of general-purpose VLMs on the task of OMR, we conducted experiments with some prompt variants. This include: (i) adding in-context learning examples, and (ii) prompting models to output MusicXML directly. We include the results in Table 4.

For in-context learning, we include three examples randomly picked from PDMX-Synth training split. Specifically, we prompt the VLMs with three rounds of conversations as follows:

```
User: Transcribe this score to ABC. <In Context Example Image 1>
AI: <Groundtruth for ICE 1>

User: Transcribe this score to ABC. <In Context Example Image 2>
AI: <Groundtruth for ICE 2>

User: Transcribe this score to ABC. <In Context Example Image 3>
AI: <Groundtruth for ICE 3>

User: Transcribe this score to ABC. <Test Image>
```

The system prompt we used to prompt VLMs to generate ABCs is the same as the prompt in appendix §A.2. The system prompt we used to prompt VLMs to generate ABCs when given in context examples is given as follows:

```
You will be given an image of a sheet of music.
Transcribe it into valid ABC 2.1 notation. Try your best to transcribe
    and make a reasonable guess if the image is not clear.
You will be given three in context examples of image-transcription pairs
Output only the ABC (no explanations), preferably inside a single ```abc
    fenced block.
IMPORTANT: NEVER give outputs like: 'Unable to transcribe from the
    provided image due to insufficient resolution/clarity.' If you can't
    tell, give your best guess.
!!!ALWAYS OUTPUT VALID ABC, DO NOT GIVE ANY ENGLISH OUTPUT. GIVE YOUR
    BEST GUESS IF YOU ARE NOT SURE!!!
```

The system prompt we used to prompt VLMs to generate MusicXML is similar:

```
You will be given an image of a sheet of music.
Transcribe it into valid MusicXML notation. Try your best to transcribe
    and make a reasonable guess if the image is not clear.
Output only the MusicXML (no explanations), preferably inside a single
    ```fenced block.
IMPORTANT: NEVER give outputs like: 'Unable to transcribe from the
    provided image due to insufficient resolution/clarity.' If you can't
    tell, give your best guess.
!!!ALWAYS OUTPUT VALID MusicXML, DO NOT GIVE ANY ENGLISH OUTPUT. GIVE
    YOUR BEST GUESS IF YOU ARE NOT SURE!!!
```

For in-context learning, we only tested it with ABC outputs because MusicXMLs are orders of magnitude more verbose than ABCs (as they include specific rendering and metadata information as well), and so within our budget we only test in-context learning with ABC notation.

Table 4: **General-purpose VLMs' ability on IMSLP Piano Scores.**

| Model | GPT-5 | | | Gemini 2.5 Pro | | |
|---|---|---|---|---|---|---|
| Output Format | ABC | ABC | MusicXML | ABC | ABC | MusicXML |
| In-Context Learning | | ✓ | | | ✓ | |
| *IMSLP Piano Scores* | | | | | | |
| TEDn | 96.7 | **74.7** | 86.6 | 87.8 | 78.6 | 80.3 |
| OMR-NED | 98.7 | **93.2** | 94.3 | 95.4 | 93.7 | 94.3 |

From Table 4, we observe that for both GPT-5 and Gemini 2.5 Pro, either in-context learning or prompting with MusicXML leads to improved performance. This finding is consistent with the observation that MusicXML resources are more prevalent than ABC notation on the internet, and that providing a few ABC examples further facilitates VLMs' understanding of this less common format.

## B.5 Focused Comparison on OpenScore String Quartets Violin Parts

We carry out a focused comparison to reduce the differences among models. We observe that SMT++ consistently produces two-staff outputs, as it is trained exclusively on piano scores. In this comparison, we manually retain only the two violin parts in the ground truth. Additionally, since SMT++ is trained on Verovio-rendered images, we also render the input images using Verovio. Furthermore, we report **kern error rates and assign empty strings to instances where Legato's output cannot be converted to **kern, further biasing the evaluation in favor of SMT++. As shown in Table 5, even under these evaluation settings extremely favorable to SMT++, Legato still outperforms it. Legato$_{small}$ underperforms SMT++ on **kern CER and SER due to conversion failure. On TEDn, which we believe is a more reasonable and comparable metric, it is far superior, though not as strong as the larger Legato model.

Table 5: **Results on OpenScore String Quartets violin parts.** Lower is better for all metrics. TEDn is the primary metric, requiring outputs to be converted to MusicXML. TEDn$_{\text{convert}}$: evaluated only on instances where SMT++ produces outputs that can be successfully converted to MusicXML; Legato outputs always converted to MusicXML successfully. OMR-NED is format agnostic, built on extraction of symbols from any format. For OMR-NED, **kern outputs are automatically corrected for syntax errors, while ABC is first converted to MusicXML (again, always successful in practice) before symbol extraction. †: 10.2% of Legato's outputs and 27.9% of Legato$_{\text{small}}$'s outputs are not convertible to **kern, so an empty prediction is used. OpenScore String Quartets is the most challenging dataset, since it has much denser score images. All metrics are explained in §5.

| Metric | SMT++ | Legato | Legato$_{\text{small}}$ |
|---|---|---|---|
| *Rendered String Quartets Violin Parts (256 pages)* | | | |
| $^{\dagger}\text{CER}_{\text{kern}}$ | 30.7 | **16.7** | 50.6 |
| $^{\dagger}\text{SER}_{\text{kern}}$ | 42.6 | **19.1** | 55.5 |
| $^{\dagger}\text{LER}_{\text{kern}}$ | 75.2 | **29.2** | 73.5 |
| TEDn | 95.0 | **7.7** | 35.0 |
| TEDn$_{\text{convert}}$ | 64.5 | **8.7** | 35.9 |
| OMR-NED | 71.2 | **15.1** | 46.2 |

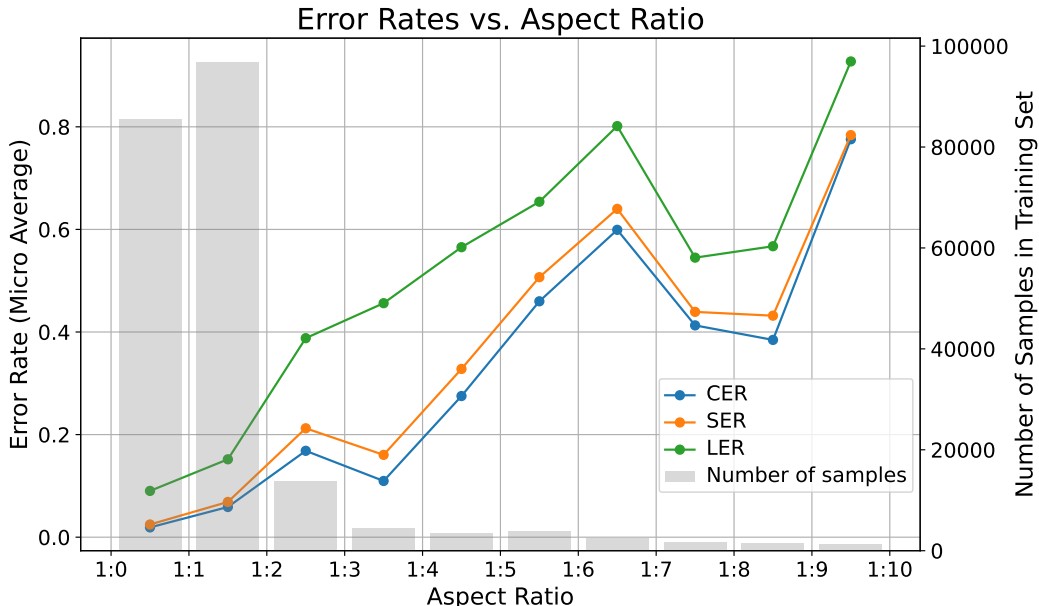

Figure 5: ABC error rates on PDMX-Synth test input with different aspect ratios. Error rates are reported by averaging over each bin. Legato is capable of recognizing multi-page scores.

### B.6 MULTI-PAGE PERFORMANCE

Figure 5 breaks down the Legato error rates (micro averaged) across different aspect ratios, showing that multi-page inputs are more challenging; it also shows they are much less frequent in the data. These inputs often result in long sequences that are truncated before reaching the decoder, limiting model performance.

### B.7 QUALITATIVE EXAMPLES

Figure 6 shows an example from IMSLP Piano Scores and output from Legato and SMT++, with errors marked in red. In this brief example—chosen as one where the TEDn$_{\text{convert}}$ score was close to the average TEDn$_{\text{convert}}$ score for each model—we can see that Legato had issues distinguishing between a 16th rest and an 8th rest, and a 32nd rest and a 16th rest, as well as one instance of

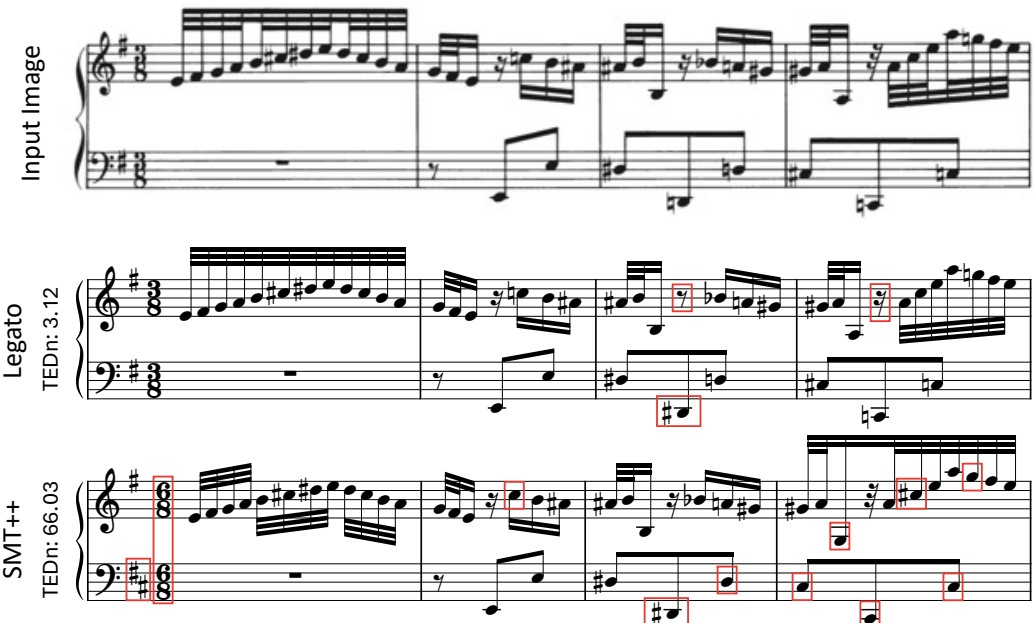

Figure 6: Example (first system of Duetto No. 1 in E minor by Bach, BWV 802) from IMSLP Piano Scores (top), with output from Legato (middle) and SMT++ (bottom). Errors are marked in red boxes.

mistaking a natural sign for a sharp. The SMT++ output incorrectly detects the time signature and the bass clef key signature, and in these 4 measures, 9 accidentals are either missing or incorrect— although we do note that in the final measure of our example, the initial C♯ is included in the key signature for that line, and the omitted sharp is likely a limitation of **kern rather than the system output.

We include more examples in Figure 7. These examples do not come from our evaluation dataset but instead were chosen as examples of famous piano compositions. We pick a scanned typesetting version that was not rendered by MuseScore, abcm2ps or Verovio. To account for the fact that SMT++ was trained only on piano data, we select only piano works for this additional example set. Additionally, we isolate a single system from the score onto a blank page for input. This provides better visualization, despite the fact that both SMT++ and Legato are capable of handling multi-system inputs.

Figure 7a shows the results on the first line of the first movement of Mozart's piano sonata K. 545. Like many beginner piano pieces, the input image is very clean, with clearly separated voices. In this particular rendition, the trill symbol is rendered in a different font than Verovio's default, and thus is unrecognizable to SMT++. In contrast, Legato can adapt to various fonts because the multiple different typesetting options in PDMX can be rendered with MuseScore, and thus are contained in the training data. Moreover, Legato correctly identifies all the slurs, while SMT++ misses them. Both **kern and ABC represent slurs using paired parentheses (), indicating that the training data for Legato also includes diverse types of slurs. This enables Legato to recognize slurs in the input image, even though it was not rendered with MuseScore, abcm2ps, or Verovio.

Figure 7b shows the results on the first system of Chopin's Op. 10 No. 4, widely regarded as an essential étude for advanced piano study. This score is more complex than the Mozart in Figure 7a, and contains multiple voices. Furthermore, the version we selected contains distortions and artifacts from the print-and-scan process, making the input more difficult for OMR. SMT++ produces an output with excessive voices, broken beaming, and inconsistent measure lengths. Some quarter rests are misaligned with eighth notes, and the clef change appears twice, making the output impossible to render with existing software. To visualize it, we manually recognized the symbols from SMT++'s output and annotated them in MuseScore. Although there are some errors in Legato's output (such as

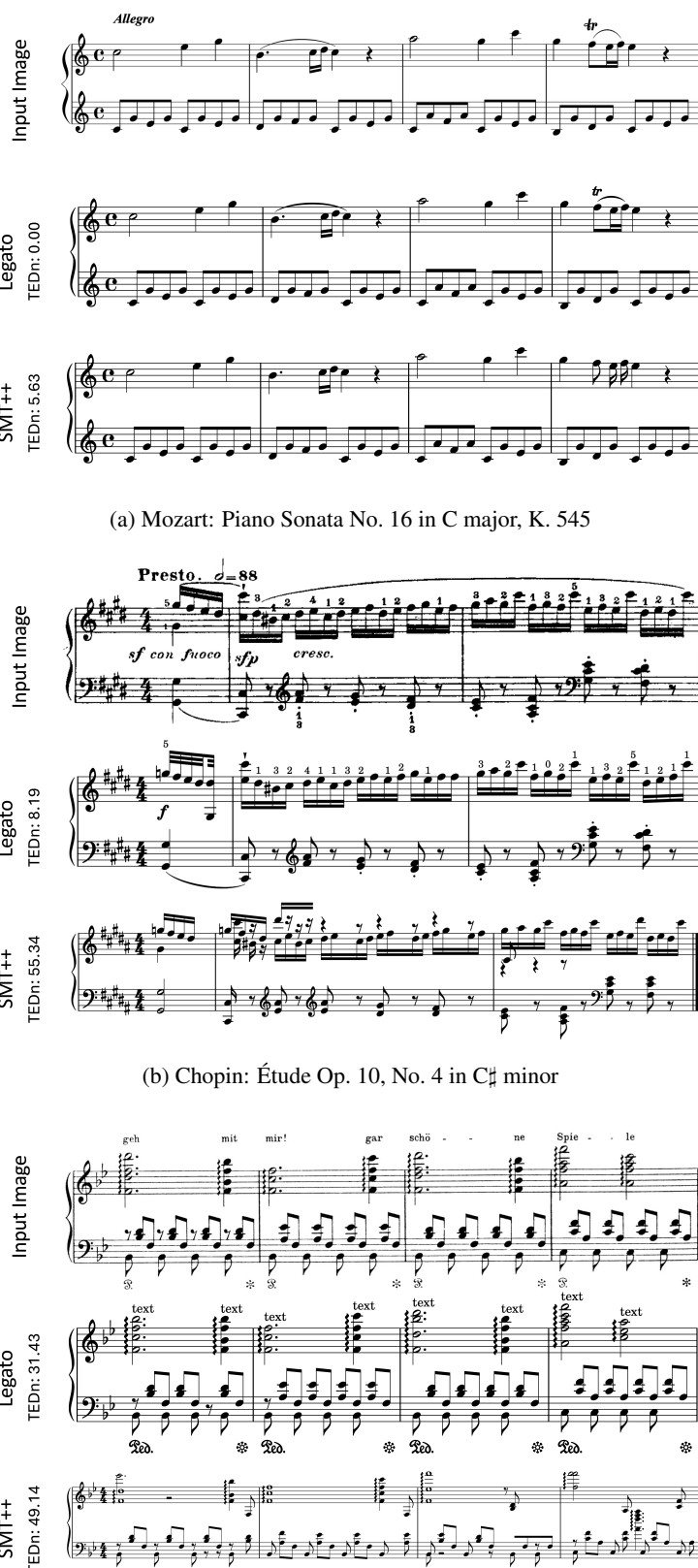

(a) Mozart: Piano Sonata No. 16 in C major, K. 545

(b) Chopin: Étude Op. 10, No. 4 in C♯ minor

(c) Liszt: Erlkönig (S. 558/4), arrangement of Schubert's D. 328

Figure 7: Qualitative examples of piano scores. Since SMT++ is trained only on piano data, we illustrate results on well-known piano works. All score images are sourced from IMSLP (not rendered by MuseScore, abcm2ps, or Verovio) and cropped to a single system for clarity, though both SMT++ and Legato can process full pages. The examples progress from easy (a), to difficult (b), to ill-formed input (c). TEDn values are reported at system-level. Since the number of errors is too large for some of the examples, we did not mark them with red boxes to preserve readability.

an ill-formed first measure), the overall quality is much better, and most fingerings and articulations are correctly detected.

Figure 7c shows a system from Liszt's *Erlkönig*. Without the context of the full score, the excerpt appears "ill-formed" because the triplet symbols are omitted. The input thus shows a quarter note in the upper staff aligned with three eighth notes in the lower staff—an unusual case absent from the training data. In addition, the eighth rests in the lower staff are missing in the second through fourth measures. While an experienced pianist can easily interpret this, the models infer literal eighth notes and rests rather than tuplets, which accounts for most of the TEDn errors. Nevertheless, Legato still outperforms SMT++ by accurately capturing the pitch of most notes and the duration of the notes in the upper staff. Moreover, Legato can also predict the placement of lyrics and pedal symbols, which lie beyond SMT++'s capabilities.

