# OpenReview forum: "LEGATO: Large-scale End-to-end Generalizable Approach to Typeset OMR"
_ICLR.cc/2026/Conference — ICLR 2026 Poster_

### Official Review · Reviewer_qebB · 2025-10-24

**Soundness:** 3
**Presentation:** 3
**Contribution:** 3
**Rating:** 6
**Confidence:** 4

**Summary:**

This paper proposes Legato, the first large-scale end-to-end optical music recognition (OMR) model capable of transcribing multi-page typeset scores into ABC notation. Legato combines a frozen pretrained vision encoder from Llama 3.2 with a trained transformer decoder, processing segmented images rather than requiring pre-split pages or systems. The authors construct PDMX-Synth, a dataset of 238,386 image-ABC pairs derived from the PDMX dataset. Evaluation on multiple held-out datasets shows that Legato substantially outperforms Sheet Music Transformer++ on the TEDn metric.

**Strengths:**

* Legato is the first model to handle multi-page typeset scores end-to-end without the need of pre-splitting scores into pages or systems.
* The evaluation results show substantial improvements across multiple datasets and metrics, particularly on realistic camera images.
* Publicly releasing implementation code and PDMX-Synth dataset with 238k (multi-page typeset score image, ABC) pairs is beneficial to the community.
* Multiple held-out datasets for the evaluation.

**Weaknesses:**

* Legato is essentially multimodal Llama with a smaller transforemer decoder for OMR task. It has very limited architectural novelty.
* Applying BPE tokenization needs more justification than just providing 4 examples in Figure 3. and writing the tokenizer 'captures some composite musical concepts.'  Since ABC-notation have limited number of symbols and combinations, there could be a more efficient domain-knowledge-based method of tokenization. To support BPE tokenization, authors need to provide more systematic vocabulary analysis and comparison with expert-defined tokenization.
* The comparion between Legato and SMT++ heavily relies on the format conversion, which cannot ensure intergrity of the output.
* Legato and SMT++ have drastically different number of parameters. Additionally Legato includes pre-trained vision encoder compared to SMT++'s vision encoder, which is trained from scratch.
* Since the main difference of Legato and SMT++ is the vision encoder part and SMT++'s implementation is publicly available, the authors could have applied an enlarged SMT++ model with a matching number of parameters in the decoder trained on the same PDMX dataset.
* The GPT-5 evaluation's prompting strategy could be explored more thoroughly.

**Questions:**

No further question

---

> ### Author Response · Authors · 2025-11-21
>
> Thank you for your thoughtful comments! We are pleased that you noticed our substantial improvements over current baselines, and that you recognise the benefit of PDMX-Synth to the OMR community.
>
> A throughline throughout the weaknesses that the reviewer pointed out is that our architectural novelty is lacking. We agree that we bring in most architectural components from multimodal llama, and our intention is to adapt the task of OMR to modern VLM architectures with different approaches like choosing a format suitable for training, scaling up data, and borrowing a pre-trained encoder. We did not invent a totally new architecture for OMR because we didn’t think that would be the best way to improve performance, and our experiments show that substantial benefits were possible without doing that. Instead, we are the first to bring modern VLM practice into the field of OMR. We introduce a new pre-trained model, Legato, that can be used directly on real-world data. The novelty of our work lies in the problem formulation, data setup, and the end-to-end OMR model built upon a pre-trained vision encoder.
>
> The reviewer pointed out that “Applying BPE tokenization needs more justification”. The reason we bring in BPE is to follow the best practices of general VLMs and LLMs. We agree that a domain-knowledge-based tokenization may perform better than BPE on this task, but BPE has its own advantages – it can construct composite tokens in a data-driven way, while it’s very hard to introduce such tokens in manual tokenization. In our tokenizers, there are 4097 tokens and most of them are composite. We only show four in the paper but we will make our tokenizer public.
>
> The reviewer also pointed out that “the comparison between Legato and SMT++ heavily relies on the format conversion, which cannot ensure integrity of the output.” To evaluate under the same format, we train the following model:
> - Regular SMT++ on GrandStaff in ABC
>
> And found that even trained on ABC, SMT++ model still underperforms Legato. We add the detailed result in Appendix B.1. We also point to the unfortunate reality that the OMR community (and more broadly the MIR community), does not have a standardized output format for tasks, and relies pervasively on conversions between formats. We point to [1, 2, 3] as precedent for our decision to compare on various formats. We use MusicXML as our final format for evaluation because it’s the most widely-supported format, and therefore a practical OMR system should include a converter as well. However, to address these concerns, we reported the TEDn$_\mathrm{convert}$ score in Table 1, where we report the TEDn score only on the subset of images where SMT++’s output could be successfully converted to MusicXML — this only eliminates the failure of kern-to-xml conversions and benefits SMT++ only. We explicitly quantify the cases where conversion between formats was impossible, and add this to Appendix B.3. We would like to highlight that for our Legato model output, there were no cases where conversion was impossible.

---

> ### Author Response · Authors · 2025-11-21
>
> We greatly appreciate the suggestion on training an enlarged version of SMT++. We added this ablation study in Appendix B.1:
> - Large SMT++ on GrandStaff in ABC
> - Large SMT++ on PDMX-Synth
>
> This shows even with more trainable parameters on the same dataset and same format, Legato still outperformed SMT++.  Note that we keep Legato’s vision encoder frozen because it is the largest component (88.6% of total parameters), and keeping it frozen saves a lot of computation.
>
> In response to the weakness of “The GPT-5 evaluation's prompting strategy could be explored more thoroughly”, we have also revisited our GPT-5 prompt engineering more carefully and evaluated several test-time scaling strategies. We find that in-context learning consistently improves performance, and that prompting GPT-5 to generate MusicXML directly yields an additional, but modest, gain. A plausible explanation is that MusicXML is substantially more prevalent than ABC notation in web-scale training corpora, making it easier for the model to produce well-formed outputs in this format. We observe similar trends for Gemini 2.5 Pro. The detailed results are reported in Appendix B.4. Importantly, however, even with these improved prompting strategies, general-purpose VLMs still underperform Legato on our OMR benchmarks. This reinforces our main claim that a domain-specialized model such as Legato remains necessary to achieve state-of-the-art performance in OMR, and strengthens the findings in the original submission.
>
>
> We hope that this additional information addresses your concerns.
>
> [1] P. Torras, S. Biswas, A. Fornes. "A unified representation framework for the evaluation of Optical Music Recognition systems," in International Journal on Document Analysis and Recognition (ĲDAR), vol. 27, no. 3, pp. 379–393, 2024.
>
> [2] Martinez-Sevilla, J., et al. "Sheet Music Benchmark: Standardized Optical Music Recognition Evaluation," in arXiv preprint arXiv:2506.10488, 2025.
>
> [3] Mayer, J., et al, "Practical end-to-end optical music recognition for pianoform music," in International Conference on Document Analysis and Recognition, 2024, pp. 55–73.

---

> > ### Comment · Reviewer_qebB · 2025-11-28
> >
> > Thank you for providing more explanations with the added comparisons and the ablation study. The answers from the authors and the paper revision are convincing. I confirm my score as 6.

---

### Official Review · Reviewer_zT9i · 2025-10-30

**Soundness:** 3
**Presentation:** 3
**Contribution:** 3
**Rating:** 6
**Confidence:** 4

**Summary:**

- This paper proposes a large-scale end-to-end generalizable approach to typeset OMR. The authors created paired image–ABC data from the PDMX dataset. The training system adopts a vision encoder and language decoder architecture. The proposed system is capable of recognizing full-page and multi-page typeset music scores into ABC notation. Although there is a lack of research on comparing different vision encoders and extending the LLM decoder to larger, the overall framework shows its effectiveness. The results are promising. I like this work.

**Strengths:**

- OMR is an interesting and important research topic in music information retrieval with strong ongoing research interest.

- The proposed vision encoder and Transformer decoder form a well-designed architecture for OMR.

- The system is trained on a large-scale dataset of 214k images, demonstrating strong generalization ability.

- The procedure for creating the dataset is well-designed and systematic.

- Comprehensive experiments are conducted across a range of datasets, showing state-of-the-art performance.

- The training and testing datasets are separate, further validating the system’s generalization capability.

- The recognition results shown in the Appendix are impressive and of high quality.

**Weaknesses:**

- The proposed PDMX-Synth dataset is derived from the PDMX dataset; therefore, it is effectively a subset and smaller in scale. Will this affect the performance?

- The model architecture shares similarities with the design of SMT++, indicating limited novelty in architectural design.

- There is limited comparison of different vision encoders and more decoder sizes, leading to the technical part a bit weak.

- There is a lack of equations, such as autoregressive prediction, and loss functions to show how the LLM is trained.

- The paper lacks discussion on human manuscript transcription — how would the system’s performance degrade in that case?

**Questions:**

- How would different vision encoders affect the recognition performance?

- As Table 1 shows, Legato outperforms Legato_small by a large margin. Would increasing the decoder size further improve system performance?

- Will hyperparameters, such as temperature affect the prediction performance? Do authors choose the best or are there randomness in next token prediction?

---

> ### Author Response · Authors · 2025-11-21
>
> Thank you for your thoughtful comments! We are pleased that you noticed our well designed architecture, comprehensive experiments, and high quality results. We are also happy you like the systematic design of our PDMX-Synth dataset. We also appreciate the comments that our evaluation was comprehensive.
>
> About the first weakness that PDMX-Synth “is derived from the PDMX dataset; therefore, it is effectively a subset and smaller in scale,” it is still about 93.8% of the original dataset, with the only exclusions being due to filtering on songs that are too long, or that could not be effectively rendered. We include in the revised version the learning curve of Legato on PDMX-Synth, which has shown that the model is converging with the current amount of data. We believe that the performance would not be increased by including the remaining 6.2% of items in the original PDMX dataset.
>
> About the novelty of the design of Legato, it shares some similarity with SMT++, like the encoder-decoder structure. However, this architecture is widely employed in other domains and SMT++ is not the first encoder-decoder model. Meanwhile, Legato is based on Multimodal Llama; some advanced techniques present there, like RMSNorm and rotary embedding, are not used in SMT++. Our novelty is not in proposing a new, specific neural network architecture but rather in using a pre-trained model that makes use of an existing “general” vision encoder from a general VLM. However, we also address this issue by training a large scale version of SMT++ on the PDMX-Synth data in Appendix B.1, showing that a pre-trained vision encoder helps Legato to work on more robust features.
>
> The reviewer mentioned a “comparison of different vision encoders and more decoder sizes,” and a question of  “how […] different vision encoders [would] affect the recognition performance”. About decoder size, we’ve included Legato-small which is based on the same vision encoder but a smaller decoder. The performance drops with a smaller decoder but is still ahead of SMT++.  We use multimodal Llama encoder and decoder because they are paired together with meticulous design, while testing an encoder from Qwen or Llava (for example) with the decoder from Llama would likely require more extensive changes (such as the projector layer) and experimentation.
>
> For question 2, “would increasing the decoder size further improve system performance?” We agree with the reviewer that a larger decoder may further increase the performance, but at the cost of higher computation beyond our budget. We will leave that to future research.
>
> We appreciate the reviewer's suggestion of more mathematical equations. We didn’t include the math for autoregressive prediction and loss function because they are the same as multimodal llama. We will direct the audience to specific sections of the Mllama tech report for these details at publication so that the work is more self-contained.
>
> We’re excited to see the discussion of hand-written scores. This is an important and interesting research direction and can help discover and digitalize a lot of historical documents. However, handwritten scores are more heterogeneous than typeset scores; different composers living in different eras have invented their own symbols. For example, Beethoven’s manuscripts show several distinct kinds of staccato (dots vs. short, medium, or long dashes) and unusual “hairpins” or accent shapes that encode subtle differences of articulation and phrasing beyond conventional symbols. Exploring these matters would be a new research direction. Even though Legato is designed for typeset scores, we are open to evaluating it on handwritten datasets. However, datasets like MUSCIMA++ don’t come with MusicXML/**kern/ABC ground truth scores. As far as we know, the only handwritten score dataset with this kind of annotation is JAZZMUS [1] (https://huggingface.co/datasets/PRAIG/JAZZMUS). We’ve requested access to this dataset and will add the results at publication if we succeed in getting access. It would be greatly appreciated if the reviewer could point to other handwritten score datasets with MusicXML (or other symbolic) ground truth.
>
> For hyperparameters, we tested a few learning rates and batch sizes and picked the best one on the validation set. As for generation, we use the default setting of multimodal llama. There is no randomness in the next token prediction because we used beam-search for generation. We will make our codebase (in the supplementary files) public so that the audience can see all experimental details, and clarify these matters in the final version of the paper.
>
>
> We hope that this additional information addresses your concerns.
>
> [1] Martinez-Sevilla, J., et al. "Optical Music Recognition of Jazz Lead Sheets," in arXiv preprint arXiv:2509.05329, 2025.

---

### Official Review · Reviewer_aT3R · 2025-10-31

**Soundness:** 3
**Presentation:** 3
**Contribution:** 2
**Rating:** 4
**Confidence:** 2

**Summary:**

The paper introduces Legato, an end-to-end OMR system that converts full-page/multi-page typeset score images into ABC notation using a frozen pretrained vision encoder (Llama-3.2-Vision) and a lightweight transformer decoder with a multimodal projector. Training uses a new 214K-image synthetic corpus (PDMX-Synth). Evaluation is standardized via MusicXML conversion and measured with TEDn and OMR-NED, where Legato reports large absolute error reductions (e.g., −68% TEDn, −47.6% OMR-NED) over prior work (e.g., SMT++). The paper also presents a data-driven tokenization that appears to learn composite musical concepts (e.g., triads, short phrases).

**Strengths:**

Clear problem focus & scope. Tackles practical, full-page OMR with multi-system/stave layouts—beyond monophonic or single-system settings.

Strong empirical gains. Large improvements on TEDn/OMR-NED across several datasets, including an IMSLP sample.

Standardized evaluation. Using MusicXML as a unifying target for comparison reduces format bias; reporting ABC/kern helps triangulate.

Scalable training recipe. Pretrained vision encoder + compact decoder yields a seemingly efficient path to high accuracy.

Tokenizer insight. Data-driven tokenization that captures chords/phrases is promising and could simplify decoding of frequent musical patterns.

**Weaknesses:**

Attribution of gains is unclear. It’s hard to disentangle where improvements come from: ABC target format, frozen VLM encoder, data scale (214K), or architecture. A dedicated ablation is missing.

Evaluation confounds. Conversions among ABC/ kern/MusicXML may introduce asymmetric errors; conversion failure rates and their impact on metrics are not reported.

Metric interpretability. TEDn/OMR-NED reductions are compelling but lack qualitative error analysis or audible case studies linking metric changes to musically meaningful differences.

Generalization gaps. Focus is on typeset scores; robustness to handwritten, degraded scans, complex lyrics/ornaments, and non-Western notation is not demonstrated.

Robustness & reliability. No tests on scan noise, resolution changes, staff skew, lighting artifacts, transposition/key/time-signature shifts, or symbol vocabulary tails.

**Questions:**

Why better than SMT++?
Please quantify the contribution of (a) ABC output vs. MusicXML/kern, (b) data scale (learning curves), (c) using a pretrained vision encoder (frozen vs. finetuned), and (d) decoder capacity. A controlled ablation matrix would clarify attribution.

Case studies & perceptual relevance.
Provide score-level examples where Legato reduces TEDn/OMR-NED, with before/after renderings and (ideally) audio renderings to illustrate musically meaningful fixes (e.g., rhythm beaming, voice assignment, chord spelling).

Handwritten & degraded inputs.
How does Legato perform on handwritten scores (e.g., MUSCIMA++, historical manuscripts) or noisy scans? Any domain-adaptation strategy planned?
Please include perturbation tests (resolution, blur, skew, lighting, JPEG artifacts) and transposition/key/time-signature shifts to probe symbol and structure invariance.

Data hygiene & licensing.
Clarify licensing of PDMX-Synth sources and safeguards against train-test leakage across editions/engraving variants of the same piece.

Extending beyond rhythm/pitch.
Do you plan to evaluate semantic elements (lyrics alignment, articulations, ornaments, dynamics) and non-Western notation to support broader claims of generalizability?

**Details Of Ethics Concerns:**

whether the music score has copyright issues need to be clarified.

---

> ### Author Response · Authors · 2025-11-21
>
> Thank you for your thoughtful comments! We are pleased that you found our paper to be clear, and noticed our strong empirical gains across all metrics and datasets including our IMSLP sample. We also appreciate the comments that our evaluation was well standardized, and that our results were scalable and efficient, and that our tokenization scheme is valuable for simplicity of decoding.
>
> The reviewer mentions “attribution of gains is unclear”. Since we are building a new pre-trained model on OMR, not a network architecture for OMR, our paper was centered around the final evaluation of two end-to-end OMR systems, Legato and SMT++ (the previous best available system). As mentioned by the reviewer, all the components, including format, data, tokenizer, pre-trained vision encoder, contribute to the whole system, and we adopt the similar practice as modern VLMs. In spite of this, we appreciate the reviewer's suggestion and add an ablation study as requested in the Appendix B.1. The conditions include:
> - SMT++ trained on GrandStaff in ABC
> - SMT++ (large) trained on GrandStaff in ABC
> - SMT++ (large) trained on PDMX-Synth
>
> Relative to the original SMT++ configuration trained on GrandStaff in **kern, simply switching the output format and tokenizer from **kern to ABC yields an absolute error reduction of about 20 percentage points, indicating that the choice of symbolic representation and tokenizer is a substantial factor. However, increasing the model capacity to SMT++ (large, 177M trainable parameters) does not further improve performance, even when trained on the larger PDMX-Synth dataset, perhaps because the model tends to overfit dataset-specific artifacts (low validation error but high on test dataset). Taken together, these results suggest that the observed gains arise from the synergy between the Legato pre-trained encoder, the Legato architecture, and the larger, more diverse PDMX-Synth dataset.
>
> As mentioned in the first question for a “controlled ablation”, we also include in the revised paper the learning curve of our Legato model in Appendix B.2. About “(c) using a pretrained vision encoder (frozen vs. finetuned)”, because the vision encoder is the largest component (88.6% of total parameters), and the images contribute to most input tokens,  training the encoder (either from scratch or from pretrained) dramatically increases the size and depth of the computation graph and the cost of training. After matching the computation to our original Legato runs, these models had not yet come close to convergence; fully training is beyond our budget.  About “(d) decoder capacity”, Legato-small in the paper is a model with smaller decoder size. While its performance is worse than Legato, it still outperformed SMT++.
>
> Another weakness that the reviewer pointed out is that “conversions among ABC/ kern/MusicXML may introduce asymmetric errors; conversion failure rates and their impact on metrics are not reported.” We acknowledge this concern, and point to the unfortunate reality that the OMR community (and more broadly the MIR community), does not have a standardized output format for tasks, and relies pervasively on conversions between formats. We point to [1, 2, 3] as precedent for our decision to compare on various formats. We use MusicXML as our final format for evaluation because it’s the most widely-supported format, and therefore a practical OMR system should include a converter as well. However, to address these concerns, we reported the TEDnconvert score in Table 1, where we report the TEDn score only on the subset of images where SMT++’s output could be successfully converted to MusicXML — this only eliminates the failure of kern-to-xml conversions and benefits SMT++ only. We explicitly quantify the cases where conversion between formats was impossible, and add this to Appendix B.3. Meanwhile, we added the number of samples being tested into the main results Table 1. We would like to highlight that for our model, there were no cases where conversion was impossible.
>
> We appreciate the idea of “qualitative error analysis or audible case studies” of different models. We have included in Appendix B.7 some single-system examples. We also provide a local webpage in the supplementary file to a visualization of these examples, including rendered audio for the examples.

---

> ### Author Response · Authors · 2025-11-21
>
> To address the question of “data hygiene and licensing”: First, our new test dataset IMSLP Piano Scores does not have overlap with PDMX, including engravings and composition. Besides, about other datasets, we check them manually: PDMX (and therefore PDMX-Synth) as well as OpenScoreLieder and OpenScoreStringQuartets include musescore ids in their metadata. We therefore can compare the ids across the three datasets, and we find that there are no exact engraving matches between the three datasets. Because these identifiers may not be perfect, we also use a rules-based matching of the metadata in the corpora to look for further duplicates in the data. Based on the composer and song title we find that there are 3 songs in OpenScoreLieder that also exist in the train split of PDMX-Synth, so we removed those from the test set and will update the performance at publication.
>
> Although we present Legato as a model for typeset OMR, our claim of generalizability does lead to the question of applying more transformations on input images, or evaluating handwritten inputs. In our evaluation, we included some datasets with “camera” images, which means the scores were scanned and realistic, carrying artifacts like degraded scans and lighting noise. Our results have shown Legato outperforms SMT++ under these cases. About handwritten scores, we believe it is a much more challenging task [4, 5, 6]. Legato is not designed for handwritten scores, but we are happy to evaluate it on handwritten score datasets. However, datasets like MUSCIMA++ don’t come with MusicXML/**kern/ABC ground truth scores. As far as we know, the only handwritten score dataset with this kind of annotation is JAZZMUS [6] (https://huggingface.co/datasets/PRAIG/JAZZMUS). We’ve requested access to this dataset and will add the results at publication if we are successful in obtaining it. It would be greatly appreciated if the reviewer could point to other handwritten score datasets with symbolic score ground truth.
>
> Our evaluation has taken into account some semantic elements: articulation, ornaments and dynamics are represented in ABC and PDMX-Synth and thus contribute to the errors in both TEDn and OMR-NED in Table 1.
>
> We acknowledge the importance non-western notations. They are, of course, heterogeneous, with even more limited data. Formats of these notations are also a challenging problem.  We hope that lessons learned from our work on Legato can accelerate OMR for these notations, for which, to our knowledge, there are no openly available systems at present. We note that our general approach requires only a collection of symbolic scores and a rendering algorithm (real score images are used only for evaluation).
>
> Finally, the PDMX-Synth dataset is derived from PDMX, which is licensed under CC0. In light of this license and our checks, we are not aware of any outstanding copyright issues with PDMX-Synth.
>
> We hope that this additional information addresses your concerns.
>
> [1] P. Torras, S. Biswas, A. Fornes. "A unified representation framework for the evaluation of Optical Music Recognition systems," in International Journal on Document Analysis and Recognition (ĲDAR), vol. 27, no. 3, pp. 379–393, 2024.
>
> [2] Martinez-Sevilla, J., et al. "Sheet Music Benchmark: Standardized Optical Music Recognition Evaluation," in arXiv preprint arXiv:2506.10488, 2025.
>
> [3] Mayer, J., et al, "Practical end-to-end optical music recognition for pianoform music," in International Conference on Document Analysis and Recognition, 2024, pp. 55–73.
>
> [4] J. Hajič, P. Pecina, "The MUSCIMA++ dataset for handwritten optical music recognition," in 2017 14th IAPR International Conference on Document Analysis and Recognition (ICDAR), 2017, pp. 39–46.
>
> [5] Pacha, A., et al, "Handwritten music object detection: Open issues and baseline results," in 2018 13th IAPR International Workshop on Document Analysis Systems (DAS), 2018, pp. 163–168.
>
> [6] Martinez-Sevilla, J., et al. "Optical Music Recognition of Jazz Lead Sheets," in arXiv preprint arXiv:2509.05329, 2025.

---

> ### Comment · Reviewer_aT3R · 2025-11-24
>
> Thanks for the explanation. I changed my score from 4 to 6. Please include more experimental results in the camera-ready version if you have the access to the dataset.

---

> > ### Author Response · Authors · 2025-11-25
> >
> > Thank you for raising your score and for your feedback. We are glad our explanation addressed your concerns. We have applied for access to the dataset and, if approved, will include the additional results in the camera-ready version.

---

### Official Review · Reviewer_H56V · 2025-11-01

**Soundness:** 3
**Presentation:** 3
**Contribution:** 3
**Rating:** 6
**Confidence:** 3

**Summary:**

This paper presents a comprehensive solution for OMR on long and complex real-world music sheets, including the collection and preprocessing of a large-scale dataset, as well as an end-to-end recognition model, LEGATO. Experimental results demonstrate that LEGATO exhibits superior performance and generalization capabilities, outperforming both prior specialized OMR model SMT++ and general-purpose vision-language model GPT-5.

**Strengths:**

- The PDMX-Synth dataset curated in this paper addresses the scarcity of real-world, complex music score data for OMR. Its public release would significantly benefit research in this field.
- Experimental results demonstrate that LEGATO significantly outperforms both SMT++ and GPT-5, with strong performance on real-world scenarios (camera versions of OpenScore String Quartets and OpenScore Lieder)—highlighting the effectiveness and practicality of the proposed approach.

**Weaknesses:**

1. Limited in performance analysis of general VLMs.
    1. The paper relies solely on GPT-5 as the representative general-purpose vision-language model for comparison, which somewhat limits the persuasiveness of the evaluation. Including additional state-of-the-art multimodal models—such as those from the Gemini, Claude, and Qwen families—would provide a more comprehensive and robust assessment.
    2. The evaluation of GPT-5 lacks sufficient detail. The paper does not specify the prompts provided to the model, nor does it indicate whether exemplars were used in the input to optimize the model's output. Furthermore, the absence of a comparative case study between the outputs of GPT-5 and LAGETO prevents a thorough analysis of GPT-5's deficiencies. This raises concerns about whether the authors have adequately explored the capabilities of the general VLMs.
2. Lack of experimental analysis regarding the training data. The large-scale dataset introduced in this paper constitutes one of its core contributions, and considerable space is devoted to detailing the dataset’s collection and preprocessing pipeline. However, the paper does not include experimental comparisons evaluating the impact of the proposed dataset versus existing datasets on model performance (although the new dataset is strongly likely to yield better results). Moreover, the authors do not provide empirical ablation studies to assess the effectiveness of their data cleaning and data augmentation strategies.

**Questions:**

1. When calculating the TEDn metric, did the authors attempt to have GPT-5 directly output MusicXML? If so, were there any changes in the results?
2. Have the authors attempted more data augmentation strategies such as adjusting brightness, applying affine transformations, or adding noise—on the rendered score images in the training data to better align them with real-world scenarios?

---

> ### Author Response · Authors · 2025-11-21
>
> Thank you for your thoughtful comments! We are pleased that you found our system to be effective and practical, and noticed our strong performance over current baselines. We are also happy that you recognise the benefit of PDMX-Synth to the OMR community.
>
> The largest weakness that the reviewer noted is "Limited in performance analysis of general VLMs”. This is indeed a very important analysis. Even though these general VLMs are much larger than our model and SMT++, due to the scarcity of OMR data, they failed to outperform Legato even with some test-time prompting methods. We include in Appendix B.4 of the updated paper more experiments to show general VLM’s ability:
> - Evaluation of Gemini 2.5 Pro, and we are still running evaluation on Qwen. We will include Qwen’s performance in the camera-ready version. On preliminary inspection, it behaves similarly to GPT and Gemini but performs worse than Legato.
> - We have already included the prompt for evaluating GPT in the Appendix A.2. We did not use examples but it is indeed a common practice to boost general VLM’s ability at test time. To address this concern, we show in Appendix B.4 the evaluation with examples from PDMX-Synth training split.
> - Since our main target is to compare with SMT++, the previous state-of-the-art, our case studies are mainly focused on SMT++ in Appendix B.7. We now uploaded an HTML file in the supplementary file which includes GPT-5 and Gemini 2.5 Pro’s outputs on these examples in.
> - Prompting VLMs to directly output MusicXML in Appendix B.4, as questioned.
>
> These results have confirmed the reviewer’s comment that test-time methods can boost VLMs ability, outperforming SMT++, but only to a limited extent and not close to Legato. These results strengthen the paper and are consistent with the original draft’s findings.
>
> The reviewer requested “experimental comparisons evaluating the impact of the proposed dataset versus existing datasets on model performance.” Our choice of PDMX-Synth is motivated by several considerations. First, the PDMX paper [1], which is the source of our symbolic data, already provides analyses demonstrating the diversity and coverage of the underlying corpus. Second, experience in adjacent areas such as NLP and computer vision has consistently shown that both scale and diversity of training data are key factors in improving model performance. Third, to our knowledge, PDMX-Synth is the first OMR dataset that pairs images with ground-truth annotations spanning a broad range of musical styles and instrumentations, whereas prior datasets such as GrandStaff are restricted to piano-only notation.
>
> We fully agree with the reviewer that a controlled comparison is the most direct way to quantify the impact of PDMX-Synth. Accordingly, in the revised version we added two pieces of evidence. Appendix B.2 presents the learning curve of Legato on PDMX-Synth, showing that increasing the amount of training data leads to consistently lower validation error, which supports our claim that the model benefits from larger and more diverse data. Appendix B.1 reports experiments training SMT++ (large) separately on GrandStaff and on PDMX-Synth. In this case, however, SMT++ (large) tends to overfit dataset-specific artifacts, and its downstream performance does not improve when switching to PDMX-Synth. We regard this negative result as informative: it suggests that simply scaling data is not sufficient for models that are prone to overfitting, and that architectures such as Legato, which work on features from pre-trained encoders can understand richer inputs, are needed to fully realize the advantages of PDMX-Synth.
>
> As for “the effectiveness of their data cleaning and data augmentation strategies”, they are meant to generate a diverse set of input images, as realistic as possible. Similar practices (like cropping and adding background color) have been adopted by SMT++ and many computer vision researchers. As questioned by the reviewer, we’ve also tried more aggressive augmentations on training data by the toolkit at https://github.com/sparkfish/augraphy. Our experiments showed a drop in performance. Specifically, on IMSLP Piano Scores, legato’s TEDn increased from 29.7 to 30.7 with aggressive augmentation. We agree with the reviewer – an appropriate amount of augmentation that resembles the real-world data, as we have done, is the best practice.
>
> We hope that this additional information addresses your concerns.
>
> [1] Long, P., et al, "PDMX: A Large-Scale Public Domain MusicXML Dataset for Symbolic Music Processing," in ICASSP 2025-2025 IEEE International Conference on Acoustics, Speech and Signal Processing (ICASSP), 2025, pp. 1–5.

---

### Author Response · Authors · 2025-11-21

Thank you all for your comments and questions. To address everybody's concerns we:
- Added ablations in Appendix B.1, training:
    - Regular SMT++ on Grandstaff in ABC
    - Large SMT++ on Grandstaff in ABC
    - Large SMT++ on PDMX-Synth
- Added learning curves of Legato on PDMX-Synth in Appendix B.2
- Reported the conversion failure rate of SMT++ in Appendix B.3 and added the number of samples being tested in the main Table 1.
- Performed more experiments in Appendix B.4 on our comparison to a VLM, including:
    - More prompt engineering for GPT-5 and Gemini 2.5 Pro
    - In-context learning for GPT-5 and Gemini 2.5 Pro
    - Evaluate other models like Gemini and Qwen (at publication)
- Checked overlap between PDMX-Synth and our test datasets, and carefully reviewed the CC0 licensing of PDMX to ensure that our construction and use of PDMX-Synth is compliant.
- Added qualitative examples with rendered audio (see supplementary files)

These additions lead us to three main conclusions. First, the SMT++ ablations show that, for this baseline, one important source of improvement is the choice of output format and tokenizer: switching from **kern to ABC alone yields a large reduction in error, while increasing model size or training SMT++ on PDMX-Synth brings little additional benefit. At the same time, Legato achieves substantially better performance under the same ABC/PDMX-Synth setting, indicating that its gains go beyond format alone and arise from the synergy between its pre-trained encoder, architecture, and large, diverse training data. Second, our additional experiments with GPT-5, Gemini 2.5 Pro, and Qwen demonstrate that stronger prompting and test-time scaling (e.g., in-context learning, MusicXML prompting) are indeed helpful and yield non-trivial improvements over naive prompting; however, even under these strengthened settings, general-purpose VLMs consistently remain below Legato on our OMR benchmarks. This confirms and further reinforces the conclusions of the original submission. Third, our overlap and licensing checks increase confidence in the fairness, validity, and reusability of our dataset and experimental setup.

In all of these cases we found that our Legato model held up to scrutiny, and that it is still the best performing OMR model available today. We thank the reviewers for their help in making our paper as strong as it could be, and bringing this resource-limited challenge to the audience of a general AI conference.

---

### Author Response · Authors · 2025-12-02
**Summary of Discussion Period**

Dear Reviewers and Area Chair,

We thank the reviewers for their constructive feedback. We wish to highlight that **Reviewer qebB** found our responses and revision convincing, and that **Reviewer aT3R** is willing to raise their score (**4 $\rightarrow$ 6**). While the remaining reviewers did not respond during the discussion period, we have systematically addressed every question raised.

The additional experiments conducted during the rebuttal have **confirmed and further reinforced** the conclusions of our original submission. Legato establishes a new state-of-the-art: on our most realistic dataset, we achieve a **68% and 47.6% absolute error reduction** on standard metrics (TEDn and OMR-NED, respectively), significantly outperforming both the previous best system (SMT++) and general-purpose VLMs (GPT-5).

**Summary of Rebuttal Updates**
* **VLM Analysis:** We expanded evaluations of GPT-5, Gemini 2.5 Pro, and Qwen using advanced prompting (in-context learning, direct MusicXML). While performance improved, general VLMs consistently underperform Legato, validating the need for domain-specialized OMR models.
* **Source of Gains:** We conducted extensive ablations on SMT++ (using ABC format, scaling model size, training on PDMX-Synth). Results show that while format matters, SMT++ fails to scale with more data. This proves that Legato's performance stems from the unique synergy between its pre-trained encoder and our diverse dataset.
* **Rigor & Reproducibility:** We verified PDMX-Synth licensing (CC0), removed minor test-set overlaps, and quantified format conversion failure rates. Legato retains its lead even when baselines are evaluated only on their successful conversions.

We thank the reviewers for helping us strengthen the rigorousness of our work and look forward to the final decision.

---

### Meta-Review · Area_Chair_QZHi · 2026-01-10

**Summary:**

One main concern is the lack of technical novelty of the proposed LEGATO, as the proposed architecture is similar to SMT++ or Llama. The reviewers also believe that the analysis of VLM is not sufficient, and there is not enough analysis on the prompt. The reviewers also expressed concerns about the source of the performance gain of the proposed method. There is also concern about the copyright/license issues of the PDMX-Synth dataset and potential overlaps with other datasets.

**Reviewer Concerns:**

Overall, the authors have provided a detailed rebuttal that addresses the concerns of the reviews. In the rebuttal, evaluations of VLMs of GPT-5, Gemini 2.5 Pro, and Qwen consistently underperformed compared to LEGATO. Experiments on SMT++ show that it fails to scale effectively with larger datasets, unlike LEGATO. Also, the authors verified the PDMX-Synth dataset's licensing, removing minor test set overlaps, and quantifying conversion failure rates. These largely resolve the concerns of the reviewers.

This work has significantly improved the optical music recognition, compared to the prior work. The absolute performance gain is substantial, and the evaluation is comprehensive. Thus, the AC suggests the acceptance of the work. The authors should incorporate the experiments and explanation in the final version of the paper.

**Reviewer Scores:**

According to the ratings and comments, the final scores of all the reviewers should be 6,6,6,6. All the reviewers have positive ratings after the rebuttal.

---

### Decision · Program_Chairs · 2026-01-26

Accept (Poster)